# Preliminary Results Regarding Sleep in a Zebrafish Model of Autism Spectrum Disorder

**DOI:** 10.3390/brainsci11050556

**Published:** 2021-04-28

**Authors:** Madalina Andreea Robea, Alin Ciobica, Alexandrina-Stefania Curpan, Gabriel Plavan, Stefan Strungaru, Radu Lefter, Mircea Nicoara

**Affiliations:** 1Department of Biology, Faculty of Biology, “Alexandru Ioan Cuza” University of Iasi, Bd. Carol I, 20A, 700505 Iasi, Romania; madalina.robea11@gmail.com (M.A.R.); andracurpan@yahoo.com (A.-S.C.); gabriel.plavan@uaic.ro (G.P.); mirmag@uaic.ro (M.N.); 2Department of Interdisciplinary Research in Science, “Alexandru Ioan Cuza” University of Iasi, Bd. Carol I Avenue, 11, 700505 Iasi, Romania; stefan.strungaru@uaic.ro; 3Center of Biomedical Research, Romanian Academy, Bd. Carol I, No 8, 700505 Iasi, Romania; radulefter@yahoo.com

**Keywords:** autism spectrum disorder, sleep, zebrafish larvae, social behavior, aggression

## Abstract

Autism spectrum disorder (ASD) is one of the most salient developmental neurological diseases and remarkable similarities have been found between humans and model animals of ASD. A common method of inducing ASD in zebrafish is by administrating valproic acid (VPA), which is an antiepileptic drug that is strongly linked with developmental defects in children. In the present study we replicated and extended the findings of VPA on social behavior in zebrafish by adding several sleep observations. Juvenile zebrafish manifested hyperactivity and an increase in ASD-like social behaviors but, interestingly, only exhibited minimal alterations in sleep. Our study confirmed that VPA can generate specific ASD symptoms, indicating that the zebrafish is an alternative model in this field of research.

## 1. Introduction

Autism spectrum disorder (ASD) is a neurodevelopmental disease which has an early onset [1,2]. The past two decades have seen a continuous and not yet explained increase in the prevalence of ASD, with an estimate of 1 in 59 children being diagnosed with ASD [3]. ASD is represented by the appearance of multiple impairments in the social domain accompanied by repetitive behaviors and restricted interests [2,3,4,5]. Apart from these symptoms, additional criteria such as anxiety, irritability, aggression, seizures, mood changes, cognitive deficits and sleep disturbances are used to diagnose autism [6]. Currently, the biological cause of autism remains unclear, but there are several risk factors, such as maternal diet, prenatal exposure to various chemical substances, age of the parents, heritability and genes linked to it [2,7,8,9,10]. It has been suggested that the cerebellum could be involved in autism since the “little brain” is one of the centers for motor coordination [11,12]. Post-mortem studies revealed a reduction in the number of Purkinje cells and neuronal cell size in the limbic system and cerebellum [12,13,14]. Some authors reported that Purkinje cells damage lead to a dysregulation of receiving the excitatory inputs and releasing GABA (gamma-aminobutyric acid), known as the principal inhibitory neurotransmitter in the central nervous system (CNS) [11,15]. Abnormalities in GABA expression were found in several brain areas in autistic people [16,17,18].

Valproic acid (VPA) has been used clinically as an antiepileptic drug (AED) to treat children and adults since the 1970s [19,20]. VPA affects the function of GABA, dopamine and serotonin neurotransmitter systems and blocks voltage-dependent sodium channels, leading to neural inactivation [18,21]. High levels of GABA in the brain were reported in clinical studies due to the quick absorption of valproate in the body [19]. Increase in synthesis, inhibition of degradation, reduction in reuptake and decrease in turnover are some of the main mechanisms implicated in GABA accumulation [21]. Furthermore, VPA can cause several adverse reactions including fatigue, headache, weight gain, somnolence and, more severely, hepatoxicity and pancreatitis [19,20,22,23,24].

Despite its regular use in epilepsy treatment, VPA is known to induce autistic characteristics in rodents and zebrafish [5,25,26,27,28,29]. Prenatal and early postnatal exposure to VPA became a new method of modeling autism in animals. Hyperactivity and repetitive/stereotypic-like behaviors were observed in rats after exposure to VPA on day 12.5 of gestation [30]. Moreover, alterations in social domain were recorded after VPA intake, leading to a decrease in time spent close to the group or in social interaction [5,25,27,29]. In addition, VPA decreases the hatching rate and locomotor activity, and induces significant genetic alterations in zebrafish embryos [28]. To our knowledge, the effects of VPA on sleep in zebrafish are not completely elucidated. Mutations of the SCN1A gene responsible for voltage-gated sodium channels function were linked to neurological disorders such as epilepsy, hemiplegic migraine and ASD [31,32]. The homozygous *scn1lab*^s552^ zebrafish mutants did not exhibit night time hyperactivity, or reduced responsiveness to dark after 25 and 250 µM VPA administration [31]. Recently, VPA-exposed larvae at 25 µM showed an increase in their locomotor activity when the lights were switched off in the first round of the dark-flash response assay, but in the end VPA-exposed larvae returned to a lower level of activity in comparison with the control group [33].

Zebrafish feature several advantages such as: rapid development, high fecundity, transparent embryos and similarity with the human genome [34,35,36]. In addition, traditional rodent models of brain disorders can readily be adapted and applied to zebrafish due to the similarities in brain morphology [34]. Moreover, the zebrafish is a suitable organism for ASD, in particular, due to similar behavioral profiles to rodents in the majority of the social status assays [37]. Zebrafish are much easier to work with and the analysis of the social deficits can be performed without many difficulties [5,26]. Due to the large number of offspring used in scientific experiments and the possibility to repeat the assay multiple times, zebrafish is preferred to the detriment of rodents.

To understand the etiology of ASD, pharmacological agents and genetic manipulation were used to investigate this disease in zebrafish. The zebrafish as a reliable novel animal model might help elucidate the ASD puzzle by bringing to light potential new findings. There are several drugs which disrupt zebrafish behavior leading to ASD-related social behavior, such as VPA, MK-801 (dizocilpine) (decreases social preference and aggression), lead pollutants and pesticides (such as chlorpyrifos and diazinon) [27,38,39]. In addition, genetic manipulation was used for discovering the mechanisms of ASD development in zebrafish. For example, the loss of shank3b function in zebrafish mutant larvae led to a reduction in locomotor activity and social interaction, besides the increase in repetitive swimming behaviors as well as decreased GABAergic and glutamatergic neurons [40,41]. Additionally, mutant larvae of cntnap2, known as a paralog for the CNTNAP2 (Contactin-Associated Protein-like 2) gene, displayed hyperactivity during the night period and a significant decrease in GABAergic cells [42].

Commonly reported by parents with ASD children are sleep disturbances [43]. Irregular sleep is likely caused by multiple factors and can have harmful effects on cognition and behavior, as well as medical impairments of the respiratory and visual systems, in children and adults [44]. Sleep disturbances have been reported in children with ASD with a range from 50 to 80%, and are manifested as difficulty initiating sleep, less overall sleep and a fragmented sleep pattern [45]. The resulting sleep deprivation leads to mood changes, loss of attention, irritability, lack of motivation and increased appetite [46]. Moreover, sleep disturbances are linked to sensory sensitivities in ASD people [47,48]. Despite the wide-reaching detrimental effects of poor sleep, dramatic improvement through various interventions has been reported in several studies [49,50,51,52]. It was shown that parent education and behavioral interventions are the most effective means in improving sleep quality and quantity [53].

Zebrafish sleep, as diurnal animals, is consistent and regulated by the circadian cycle and homeostatic mechanisms [34,35]. Similar to mammals, zebrafish possess all the neurotransmitters known to be implicated in sleep regulation, such as melatonin, hypocretins/orexins, GABA and histamine [54,55,56,57], and demonstrate an absence of voluntary movement, reversibility, increased arousal threshold, spontaneous occurrence with a circadian rhythm and homeostatic regulation [58]. Furthermore, experimental studies have shown analogous responses to sleep aids in zebrafish and humans [59].

The main goal of our study was to discover if early administration of VPA could have an impact on zebrafish sleep and to analyze how sleep parameters were changed as a result of it. Due to the fact that the diagnosis of ASD is based mainly on behavior, the validation of the VPA animal model was made through behavioral observations. Specific sleep parameters, basic locomotor activity and social interaction were quantified. Besides those aforementioned, aggression status was also measured. No significant changes could be observed for all the groups; however, these preliminary results encourage further work in analyzing the relationship between sleep and VPA as a potential ASD inductor.

## 2. Materials and Methods

### 2.1. Animal Ethics and Husbandry

All procedures were performed in strict compliance with the regulations of the Icelandic Food and Veterinary Authority and the Icelandic law (55/2013) regarding welfare of animals used for scientific purposes (460/2017). The experiment was approved through National Act no 55/2013 on Animal Welfare and National Regulation no 460/2017 License no 0004. Embryos and juvenile zebrafish (*Danio rerio*, strain WT, line AB, Zirc) were accommodated in the zebrafish facility at Reykjavik University in a zebrafish aquatic housing system (Aquatic Habitats, Apopka, FL). All zebrafish eggs were obtained by natural spawning and incubated at 28.5 °C in blue egg medium (water + methylene blue) in 2 L tanks. Eggs were randomly allocated into different tanks according to the experimental design, and maintained on a 14:10-h light-dark cycle until 6 days post-fertilization.

### 2.2. Chemical Exposure

Valproic acid (2-propylpentanoic acid, P6273-100ML, Sigma-Aldrich) was diluted in system water to a concentration of 48 µM, and administered during the first 24, 48 and 72 h post-fertilization. The VPA solution was freshly prepared before administrating it to zebrafish larvae by replacing the old medium with new one and by also adding methylene blue. The dosage was chosen according to previous studies executed in the same manner [26,27]. In the beginning of the experiment, each group had 50 individual eggs because the mortality rate can not be entirely controlled. After eliminating not fertilized or dead embryos, the animals were distributed in groups with 30 animals each for testing. After 72 h, zebrafish larvae were transferred and maintained only in system water.

### 2.3. Experimental Design

The eggs were randomly assigned into four groups (control and three VPA-treated groups) and maintained until 6 days post-fertilization. The first step of the experiment was to conduct a behavioral recording of the activity during the day and the following night. After 24 h of recording, the larvae were transferred from well-plates to tanks (20 cm × 10 cm × 10 cm, length × height × width), and maintained in system water. Starting from this point (7 days old), larvae were fed daily with larval food and raised in an incubator at 28.5 °C under ambient light control (14:10-h light-dark cycle). We chose to not feed the larvae as they can live exclusively on yolk nutrients at least until 7 days old, and also in a study conducted by Clift et al., 2014, it was observed that fed 6- and 7-day-old larvae presented alterations in swimming speed and resting compared to their unfed counterparts [60]. Zebrafish larvae were raised until 6 weeks of age. The second step of the experiment consisted of behavioral testing of juvenile zebrafish aged 6 weeks (Figure 1). Before starting recording any of the behavioral tests, we gave different acclimatization times (as it will be discussed in the following paragraphs) to the experimental chamber in order to avoid the apparatus novelty stress which might have impacted the tests’ observations.

A rectangular chamber (40 cm × 30 cm × 10 cm, length × height × width) was divided into three zones using transparent barriers to delimit the areas and to enable animals to see each other. The chambers were placed in a light- and temperature-controlled recording apparatus (14:10 light-dark cycle, lights on at 8 am, off at 10 pm, temperature 28.5 °C) at 12 pm fitted with 24 separate cameras (Ikegami, ICD-49E; Ikegami Tsushinki Co, Tokyo, Japan). The same experimental chamber was used for the aggression test for which a mirror (10 cm × 15 cm, length × height) was placed next to the tank wall as a stimulus for the experimental fish. At the end of the experiment the fish from each group were sacrificed through rapid freezing.

### 2.4. Behavioral Assessment

#### 2.4.1. Locomotor Activity

At 6 days old, all larvae groups designated for locomotor activity testing were distributed in 96-well plates containing system water. Each group consisted of 30 individual zebrafish larvae. The acclimation period consisted of 1 h before starting the 24 h recording. Locomotor activity was tracked using EthoVision XT Software (Version 11.5.2016, Noldus, Wageningen, The Netherlands) and the following parameters were analyzed: the total distance swam, average velocity and the time spent moving.

#### 2.4.2. Social Interaction Test

The social interaction test was performed in order to assess the effect of VPA on social behavior. The number of individuals for this test was: Group 1 (*n* = 30), Group 2 (*n* = 24), Group 3 (*n* = 24) and Group 4 (*n* = 22). The experimental fish were allowed to acclimate to the experimental area for 30 s followed by 20 min of recording. For this test and as well for the aggression one, an acclimation period of 30 s was chosen based on literature findings on the matter. Way and his team observed no clear significant differences in their study between a period of 30 s for acclimation and a period of 30 min, but they stated that a period of 30 min might not be long enough to observe significant behavioral differences. In any case, there are several studies assessing toxicological effects on behavior which reproduce or slightly modify the method of Gerlai, which opted for a 30 s acclimation period as well [61,62]. In order to quantify fish social preference, the experimental chamber was divided into three areas: a group zone, an individual zone and an empty zone (Figure 2). The individual zone (IZ) was divided into next to the group, starting point and next to the empty areas. The group consisted of four individual fish from the same group as the fish from the start point. The amount of time spent by the experimental fish in each area was analyzed. Depending on the time spent next to the group area, the social interaction was determined.

#### 2.4.3. Aggression

The aggression of the animals was evaluated in juvenile fish at 6 weeks of age using the mirror test and the number of individuals for each group was: Group 1 (*n* = 30), Group 2 (*n* = 24), Group 3 (*n* = 24) and Group 4 (*n* = 22) [63]. Each fish was tested in the same experimental tank used in the social interaction test to which a mirror was added. Prior to each recording, the experimental fish were allowed to acclimate for 30 s followed by recording of aggressive behavior against its own mirror image for a period of 20 min. The validation of aggressive behavior was done by measuring the time spent by the animals in the mirror zone. Thereby, the intensity of aggression was time-dependent; it went up as the time spent next to the mirror increased. The amount of time spent in each designated area (starting point, middle zone and next to mirror) was assessed (Figure 2).

#### 2.4.4. Sleep Parameter Recording

A total of 120 larval zebrafish at 6 days old were placed individually into wells of 96-well plates at 11.00 am. Larvae were assigned to four groups prior to VPA administration: Group 1 (untreated control, *n* = 30), Group 2 (24 h of VPA administration, *n* = 30), Group 3 (48 h of VPA administration, *n* = 30) and Group 4 (72 h of VPA administration, *n* = 30). Larvae were acclimatized for one hour in the recording facility prior to recording, based on previous studies analyzing sleep in zebrafish [63]. The recording was performed in a thermoregulated room at 28.0 °C, blocked from natural light, and illuminated with white and infrared light (255 lx, light-phase; 0 lx, dark-phase). Recording was initiated at 01.00 pm for a total duration of 24 h. At the end of the recording, larvae were relocated to their original tanks. Data from 10.00 pm to 08.00 am were used to analyze sleep parameters.

Prior, in-depth frame-by-frame video analysis by three independent raters resulted in establishing 1.0 mm/s as the threshold for movement. All activity that fell below that threshold was described as non-movement. Each second was considered as a whole. If the velocity at any sampling point in the second was higher than the 1.0 mm/s movement threshold, the fish was considered to be moving in that second and it was designated as a bout of mobility. If the velocities for all the sampling points in the second were below the movement threshold the fish was considered to be “not moving” in that second and designated as a bout of immobility. Next, based on prior findings [64], 6 s of consecutive immobility bouts were treated as sleep. If the time between two movements was longer than the “sleep period threshold” the fish was marked to be asleep during the time between the two movements. All other bouts were therefore considered wake bouts. According to this the shortest sleep period is 6 s. Data were exported from EthoVision XT as a text edit file and imported into a custom-built data analysis software.

### 2.5. Statistical Analysis

Movement, velocity and time spent moving were measured for behavioral analysis. Any increase in movement and maintaining it was considered hyperactivity as a result of VPA disturbance in zebrafish metabolism. Time spent moving was defined as the proportion of time spent in activity by the fish noted as active status. It was measured as an indicator for freezing bouts which could be a consequence of VPA presence. Freezing behavior is described as a complete lack of movement which can be a result of anxiety and stress, defined as stationary time in the present study [65]. The behavioral assessment data were expressed as the mean ± standard error of the mean (SEM) and analyzed by using the Student’s t-test and analysis of variance (ANOVA), followed by the Tukey HSD test. The α value was 0.05 to indicate the mean differences between the groups. The statistical analyses were made using GraphPad Prism 9 (2019) software designed and created by a Private Corporation, California. For sleep parameters the following parameters were measured: first, the percentage of time spent asleep and awake was calculated; second, the mean duration of sleep and wake bouts was calculated; third, fragmentation (number of awakenings per hour) was calculated; and fourth, swim velocity was calculated. Means were compared by MANOVA using SPSS (SPSS, Chicago, IL, USA). α value was set at 0.05 and the Bonferroni correction was used for post-hoc analyses when significant effects were found. All means are presented with their standard errors.

## 3. Results

### 3.1. VPA Administration Leads to Hyperactivity

Movement, velocity and the time spent moving were evaluated after exposure to VPA for 24, 48 and 72 h. Hyperactivity was one of the main observations recorded after 24 h of behavioral recording (Figure 3).

During the daytime, the most significant activity was recorded for Group 4 (783.7 ± 73.7 mm, *p* = 0.02) compared to Group 1 (515.4 ± 55.9 mm). In contrast, VPA administration for 48 h led to a decrease in locomotor activity for Group 3 (414.6 ± 45.6 mm, *p* = 0.001) compared to Group 4. No changes for Group 2 were found (551.1 ± 77.9 mm). Regarding the night data, the activity of Group 4 was the same as that during the day. A significant increase in night distance was swam by individuals from Group 4 (467.4 ± 47.4 mm, *p* = 0.001) compared to Group 1 (296.7 ± 30.1 mm), Group 2 (256.7 ± 34.9 mm) and Group 3 (237.4 ± 24.9 mm) (Figure 3).

Regardless of the velocity parameter, there were significant differences among the experimental groups. The highest velocity belonged to Group 4 (0.48 ± 0.03 mm/s, *p* = 0.006) compared to Group 1 (0.30 ± 0.03 mm/s), while a decrease was observed for Group 3 (0.27 ± 0.03 mm/s). The night activity tended to be the same as that from the daytime: hyperactivity for Group 4 (0.31 ± 0.02 mm/s, *p* = 0.001) compared to Group 1 (0.18 ± 0.01 mm/s). There were no changes in velocity for Group 2 (day: 0.34 ± 0.03 mm/s; night: 0.21 ± 0.01 mm/s) (Figure 4).

The time spent moving parameter (the active status) recorded significant differences between groups during the whole recording period. Group 4 (1.13 ± 0.15 s) had the most intense activity compared to the other groups in the daytime. Group 2 (0.66 ± 0.09 s) was the second most active group compared to Group 4 (*p* = 0.002), while Group 3 (0.39 ± 0.05 s) spent the lowest moving period (*p* = 0.001). When compared to the control group (0.52 ± 0.06 s), Group 4 spent clearly the most time moving (*p* = 0.001). With regard to the night period, Group 4 (0.44 ± 0.08 s, *p* = 0.001) had a significantly higher activity compared to Group 1 (0.19 ± 0.04 s) and Group 3 (0.16 ± 0.03 s) (Figure 5).

### 3.2. No Significant Effects on Sleep after VPA Administration in Zebrafish Larvae

Sleep fragmentation is characterized as repetitive short interruptions of sleep. A significant reduction in sleep fragmentation was observed for Group 4, accompanied by an increase in wake bout duration (Table 1). Significant changes were observed in movement activity during the night (*p* = 0.001) (Figure 6). Group 3 and Group 4 demonstrated opposite swim activity. Whereas Group 3, exposed to VPA for 48 h, showed a decrease in swim velocity and distance during night, Group 4, exposed to VPA for 72 h, swam substantially faster and more compared to controls (Table 1).

### 3.3. VPA Administration Leads to Impairments in Juvenile Zebrafish Social Interaction

Social interaction was strongly impaired in juvenile zebrafish. We observed a significant decrease in time spent in the group area for all three VPA-treated groups (*p* < 0.05): Group 2 (362.0 ± 28.8 s), Group 3 (336.4 ± 36.4 s) and Group 4 (217.6 ± 36.5 s), compared to Group 1 (451.6 ± 26.1 s). Time spent in the start point area by fish increased correspondingly to the period of exposure to VPA.

VPA acts as a perturbing agent, leading to impairments in the social domain of ASD individuals. Altered social interactions were the main phenotype observed in juvenile zebrafish affected by VPA. A significant decrease in sociability was observed to be time-dependent on the increase in VPA exposure. We observed a significant decrease in time spent in the closest area to the conspecific shoal (*p* < 0.05) for Group 2 (362.0 ± 28.8 s), Group 3 (336.4 ± 36.4 s) and Group 4 (217.6 ± 36.5 s), compared to Group 1 (451.6 ± 26.1 s) (Figure 7).

Furthermore, VPA treatment induced changes in the total swam distance, average velocity and time spent moving parameters (*p* < 0.05) (Figure 8). Compared to Group 1 (613.3 ± 61.2 cm), the swam distance decreased during the recording time for Group 2 (420.0 ± 37.8 cm), Group 3 (310.8 ± 40.0 cm) and Group 4 (387.0 ± 66.4 cm). The average velocity was significantly increased for Group 2 (2.26 ± 0.19 cm/s) and Group 4 (1.75 ± 0.09 cm/s) compared to Group 1 (1.57 ± 0.08 cm/s), whereas a significant decrease was observed for Group 3 (1.17 ± 0.09 cm/s). When evaluating the active status, Group 3 (81.3 ± 10.4 s) was significantly less active than Group 1 (145.4 ± 16.7 s).

### 3.4. VPA Administration Has No Effects on Juvenile Zebrafish Aggression

Exposure to VPA did not induce significant changes in the aggressive behavior of the experimental groups in the first 24 and 48 h of administration (Figure 9). The fish spent almost the same time next to the mirror or in the specific area of it (Group 1, 410.2 ± 25.5 s; Group 2, 399.2 ± 38.6 s; and Group 3, 374.0 ± 32.4 s). Instead, Group 4, who received VPA 72 h, showed an increase in time spent in mirror area (497.3 ± 38.8 s, *p* = 0.05). We quantified the aggression intensity by the time spent next to the mirror. Our results showed an increase in aggression level as the time of VPA incubation was lengthened.

## 4. Discussion

In the present study, we analyzed the effects of VPA administration on sleep parameters, along with social interaction and aggression. We obtained some preliminary results regarding early administration of 48 µM VPA on zebrafish sleep. Next, we demonstrated that embryonic exposure to 48 µM VPA could induce autistic-like symptoms in 6-day-old zebrafish larvae. In the end, we showed an increased aggression status in juvenile zebrafish larvae exposed to VPA exceeding 48 h.

The zebrafish has a well-studied sleep architecture; it possesses mechanisms for circadian regulation, specific periods for inactivity and rebound homeostasis [66]. The neurotransmitters histamine and GABA regulate wakefulness in mammals, mice and *Drosophila* [67,68]. In zebrafish, there are similar structures to mammals; for example, histamine neurons are found in the hypothalamus and noradrenaline in the locus coeruleus [69]. However, disruption of histamine signaling through genetic mutation results in subtle effects on sleep in zebrafish, as opposed to pharmacological manipulation [69]. Thus, a combined effect of histamine and GABA is evidently required to govern wakefulness [70]. Baronio et al. showed that the histamine level and the number of histaminergic neurons are significantly reduced in zebrafish larvae after 5 days of 25 µM VPA intake [33].

GABA is released by histaminergic neurons within the brain [67,71,72], and insufficient GABA release has been shown to induce hyperactivity and perturbation of the sleep cycle in mice [67]. In the present study, hyperactivity and decrease in sleep were observed for Group 4, which had the longest exposure period.

Swimming activity parameters recorded increases in values for zebrafish larvae with the highest VPA exposure period. This result is in agreement with previous studies in which larvae were exposed to 3 mM, 50 µM or 500 µM VPA in the first hours post-fertilization [73,74,75]. It indicated that prolonged VPA exposure was associated with hyperactivity during the daytime and sleep. What was not taken into account was the fact that the zebrafish does not have a center for circadian rhythm, and the majority of neurons can contribute to its regulation [76]. Therefore, as a consequence of VPA presence, the activity of neurons can be disturbed and the effects could be time-dependent.

In the present study, we recorded the highest activity after 72 h of exposure to VPA. Previous research has shown that hyperactivity can be induced in 48 h larval zebrafish after exposure to 48 µM VPA [26], and after 5 days of exposure to 50 µM VPA [74]. In a recent study, embryonic exposure initiated at 8 h and ended at day 4.5 induced hyperactivity following exposure to 5 and 50 µM VPA [27].

Studying the sociability interactions in ASD requires that animal models focus on neurobehavioral phenotypes. Studies in rodent models have demonstrated that VPA can induce deficits in the social domain. A single dose of 360 mg VPA was sufficient to induce ASD symptoms in rats [29]. Consistent with our findings, a recent zebrafish study reported that early VPA administration initiated at 8 h lasting until day 4.5 caused impaired sociability [27]. They quantified the social behavior swim distance and time spent in social contact. In contrast, we measured the time spent next to the group (stimulus for one individual) or in a specific sub-area. The VPA effect on sociability was time-dependent; time spent by zebrafish in the group area decreased as time increased.

Over the years, individuals with a neuropsychiatric disorder who exhibited aggressive and violent behaviors were treated with VPA. The efficiency of VPA is still unclear and it is not currently established whether the drug can be used to reduce aggression [77,78]. Previous reports highlight the negative action of VPA on aggression control [79]. In the present study, the effect of VPA on aggression in zebrafish was subtle as we only observed an increase in aggression for Group 4. Thus, Zimmerman et al. (2015) conducted an experiment using the same drug concentration. They demonstrated that adult zebrafish had no response to VPA intake even if it was administrated in the early hours of life [26]. Additionally, findings from Smagin et al. (2010) showed that a single dose of 100 mg/kg had no effect on aggression behavior of a male mouse, but a single dose of 300 mg/kg VPA could reduce aggression in male mice. According to them, the effect of VPA on aggression status changes proportionally with the duration of aggressivity behavior and brain neurochemical activity [80]. The data known until now present different effects of VPA which depend on the time of administration; therefore, it must be further observed.

In summary, this study expanded the information available about VPA-induced ASD phenotypes using zebrafish as a model for ASD. Our study adds some insights about zebrafish sleep after VPA intake. We report significant changes in zebrafish social behavior as a result of VPA administration in the early development stage. These findings highlight the importance of zebrafish as a tool to analyze the mechanisms underlying this condition and its relevance for understanding the causes of ASD occurrence.

## 5. Conclusions

Even if autism spectrum disorder involves a complex interaction between environment factors and genetic alterations, researchers have found new connections for prevention and treatment. In many studies which involved animal models, autism-like symptoms were developed based on the effects of environmental and genetic risk factors. In our study, we used valproic acid, known for its relevance in autism animal modeling. We recorded significant changes in zebrafish social behavior; one of the main reported autism impairments. Besides these behavioral observations, abnormalities in sleep fragmentation and wake bout duration were found. We consider that this study brought new information that might contribute significantly to this research area and opened/extended potential new methodologies with the sole purpose of helping individuals currently diagnosed with autism, as well as people working on preventing such disorders.

## Figures and Tables

**Figure 1 brainsci-11-00556-f001:**
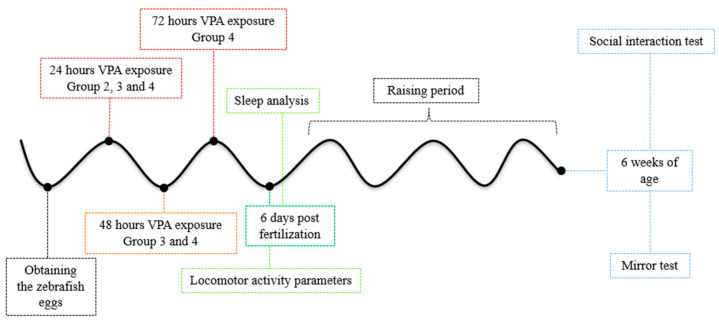
Schematic diagram of VPA exposure in zebrafish embryos. VPA administration started in the first hours post-fertilization and ended after 24 h for Group 2, 48 h for Group 3 and 72 h for Group 4. Time of evaluation and behavioral tests are mentioned.

**Figure 2 brainsci-11-00556-f002:**
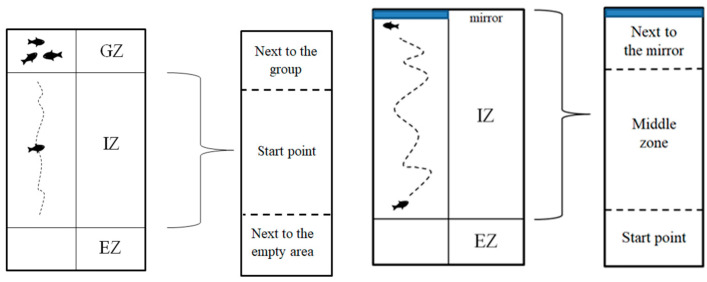
Architecture of the mazes and the main areas of it for social interaction test (**left**) and aggression test (**right**) where GZ = group zone, IZ = individual zone and EZ = empty zone.

**Figure 3 brainsci-11-00556-f003:**
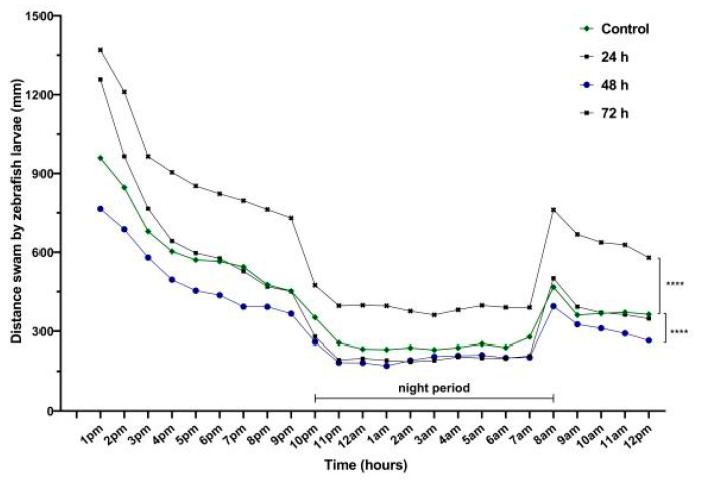
The total distance swam by 6-day-old zebrafish larvae recorded during 24 h (*n* = 30). Group 1, control; Group 2, group exposed for 24 h to VPA; Group 3, group exposed for 48 h to VPA; and Group 4, group exposed for 72 h to VPA. The data is represented as average ± SEM. Anova and Tukey post-hoc tests, *p* < 0.05 was considered to be significant. The experimental groups were compared to control group, where **** *p* < 0.0001.

**Figure 4 brainsci-11-00556-f004:**
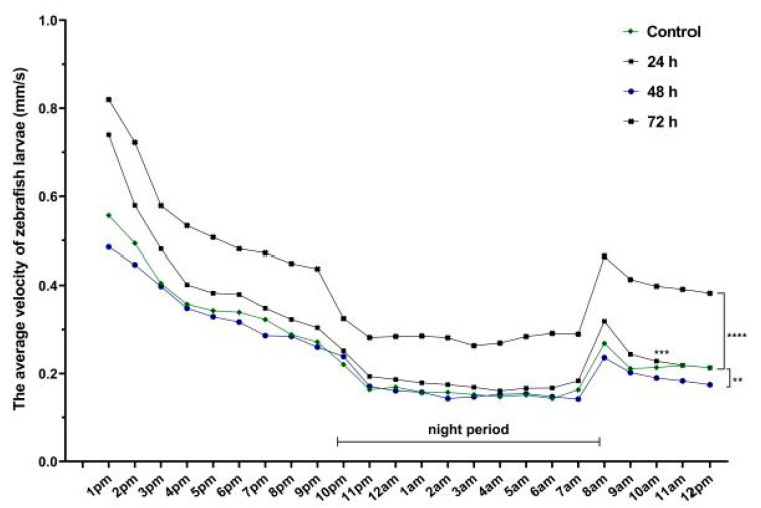
The average velocity of 6-day-old zebrafish larvae recorded during 24 h (*n* = 30). Group 1, control; Group 2, group exposed for 24 h to VPA; Group 3, group exposed for 48 h to VPA; and Group 4, group exposed for 72 h to VPA. The data is represented as average ± SEM. Anova and Tukey post-hoc tests, *p* < 0.05 was considered to be significant. The experimental groups were compared to control group, where ** *p* < 0.01; *** *p* < 0.001 and **** *p* < 0.0001.

**Figure 5 brainsci-11-00556-f005:**
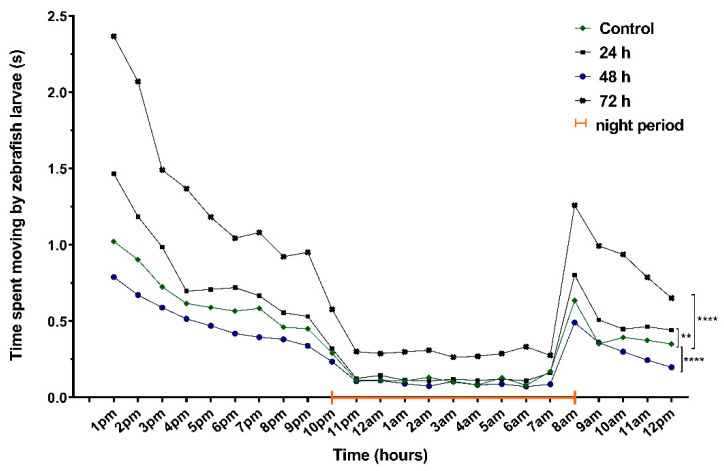
Results of time spent moving parameter (active status) of 6-day-old larvae recorded during 24 h (*n* = 30). Group 1, control; Group 2, group exposed for 24 h to VPA; Group 3, group exposed for 48 h to VPA; and Group 4, group exposed for 72 h to VPA. The data is represented as average ± SEM. Anova and Tukey post-hoc tests, *p* < 0.05 was considered to be significant. The experimental groups were compared to control group, where ** *p* < 0.01; **** *p* < 0.0001.

**Figure 6 brainsci-11-00556-f006:**
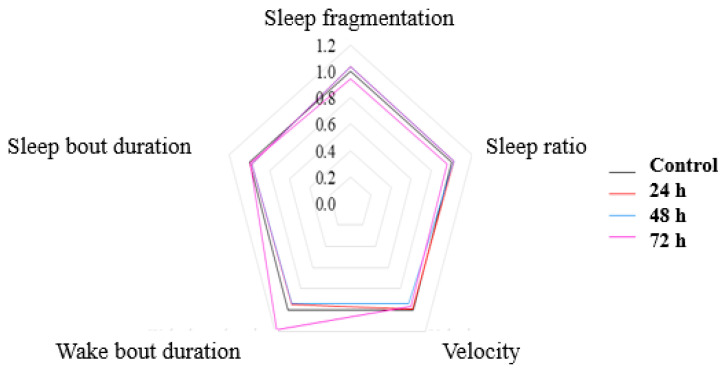
A pentagon depicting the changes in sleep parameters following VPA administration. All parameters are depicted as ratios of Group 1 (set as 1). Group 1, control; Group 2, group exposed for 24 h to VPA; Group 3, group exposed for 48 h to VPA; and Group 4, group exposed for 72 h to VPA.

**Figure 7 brainsci-11-00556-f007:**
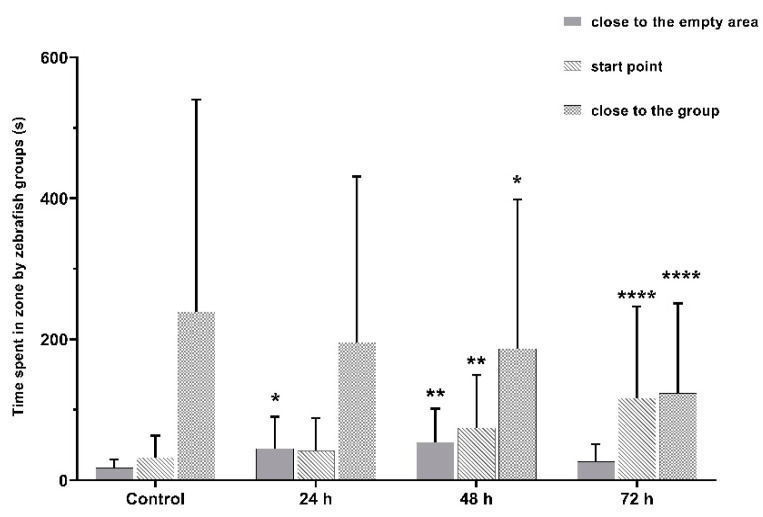
The amount of time spent by zebrafish groups during the social interaction test. Group 1, control (*n* = 30); Group 2, group exposed for 24 h to VPA (*n* = 24); Group 3, group exposed for 48 h to VPA (*n* = 24); and Group 4, group exposed for 72 h to VPA (*n* = 22). The social behavior was measured as the time spent by fish in the close to the group area. The data is represented as average ± SEM. Anova and Tukey post-hoc tests, *p* < 0.05 was considered to be significant. The experimental groups were compared to control group, where * *p* < 0.05, ** *p* < 0.01 and **** *p* < 0.0001.

**Figure 8 brainsci-11-00556-f008:**
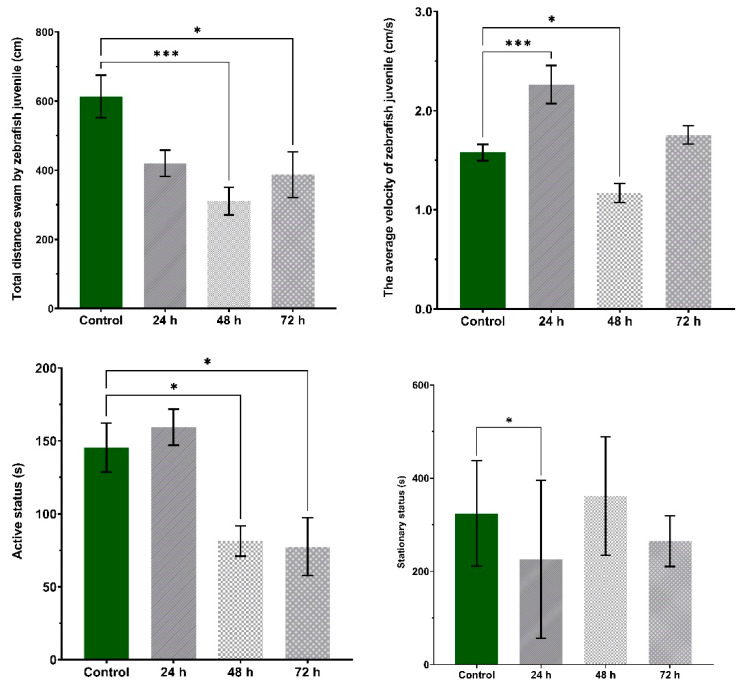
The main parameters of locomotor activity during the social interaction test. Group 1, control (*n* = 30); Group 2, group exposed for 24 h to VPA (*n* = 24); Group 3, group exposed for 48 h to VPA (*n* = 24); and Group 4, group exposed for 72 h to VPA (*n* = 22). Active status parameter represents the time spent moving by fish being active and stationary status the time spent in inactivity. The data is represented as average ± SEM. Anova and Tukey post-hoc tests, *p* < 0.05 was considered to be significant. The experimental groups were compared to control group, where * *p* < 0.05 and *** *p* < 0.001.

**Figure 9 brainsci-11-00556-f009:**
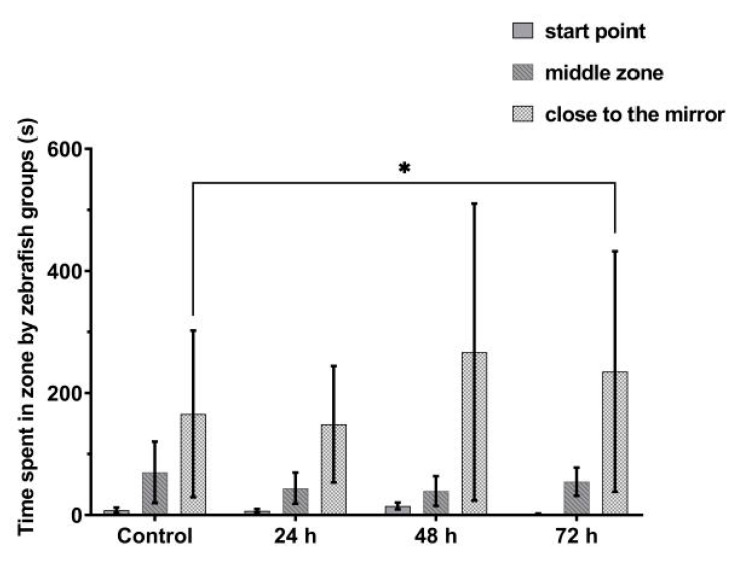
The amount of time spent by zebrafish groups during the aggression test. Group 1, control (*n* = 30); Group 2, group exposed for 24 h to VPA (*n* = 24); Group 3, group exposed for 48 h to VPA (*n* = 24); and Group 4, group exposed for 72 h to VPA (*n* = 22). The data is represented as average ± SEM. Anova and Tukey post-hoc tests, *p* < 0.05 was considered to be significant. The experimental groups were compared to control group, where * *p* < 0.05.

**Table 1 brainsci-11-00556-t001:** Sleep parameters in larval zebrafish. Group 1, control; Group 2, group exposed for 24 h to VPA; Group 3, group exposed for 48 h to VPA; and Group 4, group exposed for 72 h to VPA. The experimental groups were compared to control group and the results are represented as average ± SEM; *p* < 0.05 was considered to be significant, MANOVA and Bonferroni analysis; N.S., not significant results.

	Fragmentation	Sleep Ratio	Velocity	Total Distance Moved	Wake Bout Duration	Sleep Bout Duration
Group 1	142.19 (±2.93)	0.445 (±0.02)	0.18 (±0.01)	296.7 (±30.1)	14.094 (±1.05)	11.513 (±0.65)
Group 2	147.74 (±0.02)	0.455 (±0.02)	0.21 (±0.01)	256.7 (±34.9)	13.293 (±1.07)	11.208 (±0.67)
Group 3	147.67 (±3.03)	0.456 (±0.02)	0.18 (±0.01)	237.4 (±24.9)	13.261 (±1.09)	11.237 (±0.68)
Group 4	133.78 (±2.75)	0.426 (±0.02)	0.31 (±0.02)	467.4 (±47.4)	16.701 (±0.99)	11.379 (±0.61)
*p*-value	0.002	N.S.	0.001	0.001	N.S.	N.S.

## Data Availability

All data used in this experiment are available on request.

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
