# Peer review of "Preliminary Results Regarding Sleep in a Zebrafish Model of Autism Spectrum Disorder"

_brainsci, 2021, doi:10.3390/brainsci11050556_

Round 1

Reviewer 1 Report

Manuscript ID: brainsci-1179012

Title: Preliminary results regarding sleep in a zebrafish model of autism spectrum disorder

General Comments:

In this manuscript the authors discuss the use of zebrafish to model ASD behaviors as well as the potential behavioral effects that occur do to VPA-induced ASD in zebrafish and their implications on sleep. The manuscript is well written and easy to follow, and uses good n values for a behavioral study. One shortcoming of the study is the authors do not discuss some of the drawbacks of using behavioral tests (ie. Low repeatability, high variability in response etc.) this should be included to better contextualize the results.  The minor comments I have listed below should be addressed prior to publication.

Specific Comments:

Line 25 – Is it actually an increase in prevalence or is it just an increase in the frequency of diagnosis? How can you delineate those two things?

Line 59 – List the specific neurological disorders it is linked to.

Line 136 – What was the mortality rate for your control eggs? If mortality is overall quite high this could have an impact on how your results are interpreted

Line 139 – Was a daily renewal used? Did you analytically validate your exposure concentrations? How was the dosage used chosen?

Figure 1. Very helpful graphic, simple and easy to follow.

Line 172 – How much time was given for the larvae to acclimate?

Line 186 – The authors state that the vast majority of studies use a 30s acclimation period, however they only cite two studies from the same author, which clearly does not represent the ‘vast majority of studies’. This needs to be rephrased, as this statement is inaccurate.

Line 211: Reads “11.00 am”, punctuation error.

Line 237: Awkwardly phrased, authors should revise.

Figure 3. Instead of just listing “group” the authors should include exposure time in the legend, it is more useful to the reader. Same for the results section and figure 4. Figures also have very poor resolution (though this could be an artifact of the draft we are given).

Figure 8. The two panels have error bars in different formats, these figures should be edited to appear in the same format.

Line 479- You did not quantify GABA expression in any way so this statement is a reach and should be removed.

Author Response

Dear Editor,

            We are writing in regards to the major revision we received for our manuscript entitled ”Preliminary results regarding sleep in a zebrafish model of autism spectrum disorder”, and authored by Madalina-Andreea Robea, Alin Stelian Ciobica, Alexandra Curpan, Gabriel Plavan, Stefan-Adrian Strungaru and Mircea Nicoara.

We will hereby respond to the reviewers’ comments by detailing the changes we made on the manuscript.

All the modifications through the manuscript are highlighted in green.

#Reviewer 1

Specific Comments:

  • Line 25 – Is it actually an increase in prevalence or is it just an increase in the frequency of diagnosis? How can you delineate those two things?

            This is a good question, thank you. From our point of view autism prevalence and diagnose are strongly linked since autism is better diagnosed after the last version of DSM-V (2013). Without an useful tool as DMS-V, we could not help the people with autism because the specialists does not have no way of knowing if it is autism or no. So, the prevalence of this disorder is directly influenced by the number of people which are identified with one of the three main levels of severity and, of course, taking into account the genetic predisposition and environmental risk factors. Thank you!

  • Line 59 – List the specific neurological disorders it is linked to.

            As the reviewer suggested, we added in the manuscript the specific neurological disorders linked to SCN1A mutations like: ”Mutations of SCN1A gene responsible with voltage-gated sodium channels functioning were linked to neurologic disorders as epilepsy, hemiplegic migraine and ASD [31], [32].

  • Line 136 – What was the mortality rate for your control eggs? If mortality is overall quite high this could have an impact on how your results are interpreted.

            At the beginning of the study we kept 50 eggs for each group and in the end we used only 30. The mortality rate was not significant since the embryos are highly sensitive in the first hours of life and the number of dead or unfertilized embryos was too small (around 10-12 embryos).

  • Line 139 – Was a daily renewal used? Did you analytically validate your exposure concentrations? How was the dosage used chosen?

The VPA solution was daily renewed and we choose this concentration based on Zimmerman’s study (“Embryological exposure to valproic acid induces social interaction deficits in zebrafish (Danio rerio): A developmental behavior analysis” 2015, DOI: 10.1016/j.ntt.2015.10.002) and Chen’s study (“Developmental and behavioral alterations in zebrafish embryonically exposed to valproic acid (VPA): An aquatic model for autism” 2018, DOI: 10.1016/j.ntt.2018.01.002) as we already mentioned it in the chemical exposure paragraph “The dosage was chosen according to previous studies executed in the same manner [26], [27].”.

  • Figure 1. Very helpful graphic, simple and easy to follow.

            Thank you for your appreciation.

  • Line 172 – How much time was given for the larvae to acclimate?

The 6 day old larvae were acclimated in the experimental chamber for one hour before the recording. Also, we added this in the manuscript: “The acclimation period consisted in 1 hour before starting the 24 hours recording.”.

  • Line 186 – The authors state that the vast majority of studies use a 30s acclimation period, however they only cite two studies from the same author, which clearly does not represent the ‘vast majority of studies’. This needs to be rephrased, as this statement is inaccurate.

                As the reviewer indicated we changed the phrase: “In any case, there are several studies assessing toxicological effects on behavior reproduce or slightly modify the method of Gerlai which opted for a 30 s acclimate period as well [61], [62].” instead of “In any case the vast majority of studies assessing toxicological effects on behavior reproduce or slightly modify the method of Gerlai which opted for a 30 s acclimate period as well [61], [62].” There is another study of Serra et al. 1999 (doi.org/10.1590/S0100-879X1999001200016) which used the same period for acclimation and not to mention our own studies in which we have successfully used 30s acclimation before the trials.

  • Line 211: Reads “11.00 am”, punctuation error.

            We corrected the mistake as the reviewer indicated.

  • Line 237: Awkwardly phrased, authors should revise.

            We modify the phrase as it was suggested by the reviewer: “Time spent moving was defined as proportion of time spent in activity by the fish noted as active status. It was measured as an indicator for freezing bouts which could be a consequence of VPA presence.” instead of “Time spent moving was defined as proportion of time spent in activity (active status) and measured to observe if VPA can induce or not freezing bouts.”.

  • Figure 3. Instead of just listing “group” the authors should include exposure time in the legend, it is more useful to the reader. Same for the results section and figure 4. Figures also have very poor resolution (though this could be an artifact of the draft we are given).

We added in the figure text informations about the exposure time for each group to avoid filling the graphs with more text (eg. “Figure 3. The total distance swam by 6 day old zebrafish larvae recorded during 24 hours (n=30). Group 1: control, Group 2: group exposed for 24 h to VPA, group 3: Group exposed for 48 h to VPA and group 4: Group exposed for 72 h to VPA. The data is represented as average ± SEM. Anova and Tukey post-hoc tests, p< 0.05 was considered to be significant. The experimental groups were compared to control group, where ****p< 0.0001.”).

  • Figure 8. The two panels have error bars in different formats, these figures should be edited to appear in the same format.

            We modify the graphs to have the same format as it was recommended.

  • Line 479- You did not quantify GABA expression in any way so this statement is a reach and should be removed.

This phrase “We can assume that VPA administration could disturb the activity of histaminergic neurons and consequently to an insufficient release of GABA, but more studies are needed.” was just a supposition of ours based on the literature informations and it was removed as the reviewer kindly indicated.

Reviewer 2 Report

Robea et al has modified the discussion section from the previous revised version. In the earlier revised version, the authors has responded to my concerns. I dont have any further concern or suggestion.

Author Response

Thank you !

Round 2

Reviewer 1 Report

The authors failed to address the majority of the comments from my previous review. For this reason I do not feel this manuscript is fit for publication at this time.

Author Response

Please accept our apologies for not responding properly to all your suggestions in our last letter. This was unacceptable from our side. Thank you for your understanding.

Specific Comments:

  • Line 25 – Is it actually an increase in prevalence or is it just an increase in the frequency of diagnosis? How can you delineate those two things?

We hope we did answer correctly this time to this observation. We have changed the line to:

“The past two decades have seen a continuous and not yet explained increase in the prevalence of ASD, with an estimate of 1 in 588 children being diagnosed with ASD [3]“

Indeed, this a subject that has been debated extensively in the recent decade; the improvements and changes in the diagnosis of autism provided by the standardized criteria of the DSM (currently V since 2013) clearly play a part in the increased trend of autism cases – and some studies would conclude that while “the true increase in incidence is not known“, the observed increase due to better diagnosis is factual (1,2). In recent meta-analyses, (3,4) authors highlight the variations in ASD prevalence depending on the methodological diagnostic differences (amongst others improvement in living conditions and healthcare professionals), thus ASD prevalence ranges from 0.06% in Iran to 2.64% in Korea (3). (5) calls this an apparent increase in prevalence and provides a detailed list on the challenges to identify the actual prevalence evolution during the last decades. Currently, delineating the actual prevalence from the apparent one remains an unanswered problem with only several hypotheses being suggested (exposure to environmental injury, genetics etc); but, whether apparent or not, this, unfortunately, does not change the fact that the prevalence has been (described as) increasing more and more lately (with over 175% in 2016 higher than in 2000 and 2002 (6)), which is what we stated in line 25, without referring to a vaster subject, beyond the scope of the paper. (5), citing 3 prevalence studies (amongst many others in the literature) – chosen for their adequate methodology -, expresses concern for the confirmed high prevalence figure for ASD (whether apparent or not) in all of the studies, and on this account, “ASD should no longer be considered a rare disorder“(5). As a final note, we would like to refer to a recent hypothesis of (7) on the subject, in which the massive use of online technology is proposed to be one of the factors contributing to the increased ASD prevalence.

1 - Croen, L.A., Grether, J.K., Hoogstrate, J. et al. The Changing Prevalence of Autism in California. J Autism Dev Disord 32, 207–215 (2002)

2- Marissa King, Peter Bearman, Diagnostic change and the increased prevalence of autism, International Journal of Epidemiology, Volume 38, Issue 5, October 2009, Pages 1224–1234

3- Qiu S, Lu Y, Li Y, Shi J, Cui H, Gu Y, Li Y, Zhong W, Zhu X, Liu Y, Cheng Y, Liu Y, Qiao Y. Prevalence of autism spectrum disorder in Asia: A systematic review and meta-analysis. Psychiatry Res. 2020 Feb;284:112679. doi: 10.1016/j.psychres.2019.112679. Epub 2019 Nov 5. PMID: 31735373.

4- Chiarotti, F., & Venerosi, A. (2020). Epidemiology of Autism Spectrum Disorders: A Review of Worldwide Prevalence Estimates Since 2014. Brain Sciences, 10(5), 274. 

5- Charman, T. (2002). The prevalence of autism spectrum disorders. European Child & Adolescent Psychiatry, 11(6), 249–256. doi:10.1007/s00787-002-0297-8 

6- https://doi.org/10.1002/cbl.30470 

7 - D.M. Barros, Online dating, reproductive success and the rise autism spectrum disorder prevalence, Medical Hypotheses, Volume 140, 2020, 109679, ISSN 0306-9877, https://doi.org/10.1016/j.mehy.2020.109679.

  • Line 59 – List the specific neurological disorders it is linked to.

            As the reviewer suggested, we added in the manuscript the specific neurological disorders linked to SCN1A mutations like: ”Mutations of SCN1A gene responsible with voltage-gated sodium channels functioning were linked to neurologic disorders as epilepsy, hemiplegic migraine and ASD [31], [32].

  • Line 136 – What was the mortality rate for your control eggs? If mortality is overall quite high this could have an impact on how your results are interpreted.

            At the beginning of the study we kept 50 eggs for each group and in the end we used only 30. The mortality rate was not significant since the embryos are highly sensitive in the first hours of life and the number of dead or unfertilized embryos was too small (around 10-12 embryos).

  • Line 139 – Was a daily renewal used? Did you analytically validate your exposure concentrations? How was the dosage used chosen?

The VPA solution was daily renewed and we choose this concentration based on Zimmerman’s study (“Embryological exposure to valproic acid induces social interaction deficits in zebrafish (Danio rerio): A developmental behavior analysis” 2015, DOI: 10.1016/j.ntt.2015.10.002) and Chen’s study (“Developmental and behavioral alterations in zebrafish embryonically exposed to valproic acid (VPA): An aquatic model for autism” 2018, DOI: 10.1016/j.ntt.2018.01.002) as we already mentioned it in the chemical exposure paragraph “The dosage was chosen according to previous studies executed in the same manner [26], [27].”.

  • Figure 1. Very helpful graphic, simple and easy to follow.

            Thank you for your appreciation.

  • Line 172 – How much time was given for the larvae to acclimate?

The 6 day old larvae were acclimated in the experimental chamber for one hour before the recording. Also, we added this in the manuscript: “The acclimation period consisted in 1 hour before starting the 24 hours recording.”.

  • Line 186 – The authors state that the vast majority of studies use a 30s acclimation period, however they only cite two studies from the same author, which clearly does not represent the ‘vast majority of studies’. This needs to be rephrased, as this statement is inaccurate.

                As the reviewer indicated we changed the phrase: “In any case, there are several studies assessing toxicological effects on behavior reproduce or slightly modify the method of Gerlai which opted for a 30 s acclimate period as well [61], [62].” instead of “In any case the vast majority of studies assessing toxicological effects on behavior reproduce or slightly modify the method of Gerlai which opted for a 30 s acclimate period as well [61], [62].” There is another study of Serra et al. 1999 (doi.org/10.1590/S0100-879X1999001200016) which used the same period for acclimation and not to mention our own studies in which we have successfully used 30s acclimation before the trials.

  • Line 211: Reads “11.00 am”, punctuation error.

            We corrected the mistake as the reviewer indicated.

  • Line 237: Awkwardly phrased, authors should revise.

            We modify the phrase as it was suggested by the reviewer: “Time spent moving was defined as proportion of time spent in activity by the fish noted as active status. It was measured as an indicator for freezing bouts which could be a consequence of VPA presence.” instead of “Time spent moving was defined as proportion of time spent in activity (active status) and measured to observe if VPA can induce or not freezing bouts.”.

  • Figure 3. Instead of just listing “group” the authors should include exposure time in the legend, it is more useful to the reader. Same for the results section and figure 4. Figures also have very poor resolution (though this could be an artifact of the draft we are given).

We are sorry for not replying to this earlier. We added in the figure text informations about the exposure time for each group (eg. “Figure 3. The total distance swam by 6 day old zebrafish larvae recorded during 24 hours (n=30). Group 1: control, Group 2: group exposed for 24 h to VPA, group 3: Group exposed for 48 h to VPA and group 4: Group exposed for 72 h to VPA. The data is represented as average ± SEM. Anova and Tukey post-hoc tests, p< 0.05 was considered to be significant. The experimental groups were compared to control group, where ****p< 0.0001.”) and also we adjusted the legend with exposure time. Regarding the resolution of the figures we improved them as the reviewer kindly suggested.

  • Figure 8. The two panels have error bars in different formats, these figures should be edited to appear in the same format.

We modify the quality of the graphs to have the same format as it was recommended and avoiding a poor data representation. Thank you again.

  • Line 479- You did not quantify GABA expression in any way so this statement is a reach and should be removed.

This phrase “We can assume that VPA administration could disturb the activity of histaminergic neurons and consequently to an insufficient release of GABA, but more studies are needed.” was just a supposition of ours based on the literature informations and it was removed as the reviewer kindly indicated.

This manuscript is a resubmission of an earlier submission. The following is a list of the peer review reports and author responses from that submission.

Round 1

Reviewer 1 Report

Robea et al has studied the impact of valproic acid (VPA) on zebrafish sleep. The authors concluded that juvenile zebrafish manifested hyperactivity and increase in ASD-like social behaviors but, only exhibit a minimal alteration in sleep. Few previous studies have reported that VPA treatment increased sleep duration in children during treatment. Findings from this study is interesting, it will add value to use zebrafish as an alternative model to study ASD as well as pharmacological studies. However, I have certain concerns which I would like to bring to the authors attention.

  1. Since in the title of the manuscript authors mentioned zebrafish as a model for ASD; please add some literature related to it in the introduction. Lots of studies have been performed to replicate ASD related phenotype in zebrafish by target gene mutation (including ASD associated sleep caused by cntnap2 mutation). As the main strength of the paper in behavior analysis try to refer few earlier ASD relate zebrafish behavior studies.
  2. It seems the zebrafish husbandry is having multiple strains. So, for statistical analyses, were zebrafish allocated to experimental groups without randomization and blinding?
  3. Sometimes VPA exposure cause morphological deformities in zebrafish. If I understand correctly, 50 embryos were exposed to VPA but only 30 were selected for the downstream analysis. How those 30 were selected? Random selection or based on morphology?
  4. Can the author mention the number of n in each group used for the locomotor activity, social interaction test and aggression test like they did for the sleep parameter recoding in the method section. Its written with the Figure legends; but better to have the information in method section also.
  5. I am confused with section 3.1 of Results. I failed to correlate the figures with the writings. It might be swap in figures? But I will not predict anything without getting clarifications from the authors. In Figure 4 legend says “The average velocity recorded during the day”; but I cannot find any velocity information in the figure.
  6. In Figure 5 it appears maximum movement was observed at 12:00-1:00 pm duration. Even morning light flash at 8:00 am fail to restore the same level of movement. Can the author explain the reason behind this? Also, was food served during this 24 hr observation? If yes how were the movement during those intervals?
  7. In figure 7 Group 4 appears to spend less time compared to Group 1 in both active as well as stationary state. Since the fish of all group were observed for same amount of time, is there any other state apart from active and stationary where Group 4 fish are spending time?
  8. Figure 2 which explains social interaction test, needs some clarification. While performing social interaction test whether the group of fish were placed in different tank or same tank separated by transparent chamber. Looking at Figure 2 I failed to understand when the fish was considered to be in the group zone. How the zones were divided 1:2:1 or something else.
  9. Earlier report suggests VPA exposure upto 48 hpf altered anxiety in juvernile and adult zebrafish. Can the author report what the impact of 72 hr exposure on zebrafish anxiety?

The main strength of this paper is sleep analysis caused by VPA administration for which method section is well written and results were presented nicely.

Author Response

Reviewer  1:

”I have certain concerns which I would like to bring to the authors attention:

  1. Since in the title of the manuscript authors mentioned zebrafish as a model for ASD; please add some literature related to it in the introduction. Lots of studies have been performed to replicate ASD related phenotype in zebrafish by target gene mutation (including ASD associated sleep caused by cntnap2 mutation). As the main strength of the paper in behavior analysis try to refer few earlier ASD relate zebrafish behavior studies.

 - We added a new paragraph in the introduction describing zebrafish as an ASD model like: „ To understand the etiology of ASD, pharmacological agents and genetic manipulation were used to investigate it in zebrafish. Considered model system, zebrafish can bright new findings in ASD puzzle. There are several drugs which disrupt zebrafish behavior leading to ASD-related social behavior such as VPA, MK-801, ketamine, and phencyclidine [27], [38]. In addition, genetic manipulation was used for discovering the mechanisms of ASD development in zebrafish. For example, the loss of shank3b function in zebrafish mutant larvae conducted to a reduction in locomotor activity and social interaction, beside the increase in repetitive swimming behaviors [39]. Also, mutant larvae of cntnap2, known as paralog for CNTNAP2 (Contactin Associated Protein-like 2) gene, displayed hyperactivity during the night period and a significant decrease in GABAergic cells [40].”

  1. It seems the zebrafish husbandry is having multiple strains. So, for statistical analyses, were zebrafish allocated to experimental groups without randomization and blinding?

- We only worked with strain WT, line AB zebrafish.

  1. Sometimes VPA exposure cause morphological deformities in zebrafish. If I understand correctly, 50 embryos were exposed to VPA but only 30 were selected for the downstream analysis. How those 30 were selected? Random selection or based on morphology?

- Not all the embryos had visible malformations; we added first the embryos who presented malformations and completed the group by random selection.

  1. Can the author mention the number of n in each group used for the locomotor activity, social interaction test and aggression test like they did for the sleep parameter recoding in the method section. Its written with the Figure legends; but better to have the information in method section also.

- We added the n number ”Each group had 30 individuals” in the locomotor activity section,

”The number of individuals for this test was: Group 1 (n=30), Group 2 (n=24), Group 3 (n=24) and Group 4 (n=22)” for social interaction test and ”the number of individuals for each group was: Group 1 (n=30), Group 2 (n=24), Group 3 (n=24) and Group 4 (n=22)” for aggression test in the method section as the reviewer kindly suggested.

  1. I am confused with section 3.1 of Results. I failed to correlate the figures with the writings. It might be swap in figures? But I will not predict anything without getting clarifications from the authors. In Figure 4 legend says “The average velocity recorded during the day”; but I cannot find any velocity information in the figure.

- We checked again the manuscript and there was an overlap problem. We added the graphs for the average velocity during the day and night.

  1. In Figure 5 it appears maximum movement was observed at 12:00-1:00 pm duration. Even morning light flash at 8:00 am fail to restore the same level of movement. Can the author explain the reason behind this? Also, was food served during this 24 hr observation? If yes how were the movement during those intervals?

- The larvae were not fed until 7 dpf; therefore, alterations in movement due to the feeding is not a factor.

  1. In figure 7 Group 4 appears to spend less time compared to Group 1 in both active as well as stationary state. Since the fish of all group were observed for same amount of time, is there any other state apart from active and stationary where Group 4 fish are spending time?

- Significant activity was recorded for Group 4 during the social interaction test where we measured the time spent next to the group. As it can be seen in the Figure 8, Group 4 exhibited an increase in the time spent in the start point compared to Group 1.

  1. Figure 2 which explains social interaction test, needs some clarification. While performing social interaction test whether the group of fish were placed in different tank or same tank separated by transparent chamber. Looking at Figure 2 I failed to understand when the fish was considered to be in the group zone. How the zones were divided 1:2:1 or something else.

- We described the experimental chamber in the section named Experimental design as:”A rectangular chamber (40 cm × 30 cm × 10 cm, length × height × width) was divided into three zones using transparent barriers to delimit the areas and enabling animals to see each other.”  which means that all the fish were in the same place and the zones division ”In order to quantify fish social preference, the experimental chamber was divided in three areas: a group zone, an individual zone and an empty zone”. We added how we quantified sociability from the individual zone: ”The individual zone (IZ) was divided in next to the group, start point and next to the empty areas”. Regarding the division of the zones, we adjusted the individual zone to be double compared to the group zone and to the empty zone because we wanted to allow fish to choose where it swims: towards the stimulus (group) or not. We also modified the Figure 2 for social interaction test diagram.

  1. Earlier report suggests VPA exposure up to 48 hpf altered anxiety in juvenile and adult zebrafish. Can the author report what the impact of 72 hr exposure on zebrafish anxiety?

- Our study has been focused on studying sleep and additionaly we quantified sociability and aggression. Unfortunately, we can not report any signs of anxiety for zebrafish 72 hr VPA exposure.”

Reviewer 2 Report

Manuscript ID: biology-870273

Title: Sleep in a Zebrafish Model of Autism Spectrum Disorder

General Comments:

Overall, this is an interesting paper exploring the impacts of pharmacologically induced autism like behaviors in zebrafish on sleep and activity. The paper is relatively well written with only a few minor grammatical errors throughout and is a good fit for the journal. There are, however, some methodological short comings that limit the usefulness of the study. For the aggression and social behavioral assays the acclimation periods are much too brief, and novel tank effects likely impacted or completely altered the behavioral responses the authors were trying to observe. For behavioral tests with zebrafish, a 20 minute acclimation period is recommended. The authors also spend a large portion of both the introduction and discussion discussing the molecular mechanisms behind VPA exposure even though these mechanisms are not at all explored in the study. The authors need to re-write the discussion and frame it around the behaviors included in the study.

Specific Comments:

Line 57 – the function of scn1lab should be described to provide context

Line 147 – with only 30s of acclimation could the ‘novel tank’ not impact the behavioral response? How did you control for the novel tank effect?

Line 164 - How did you distinguish between aggressive and social behaviors in the mirror? Being in the same zone as the mirror does not necessarily indicate an aggression response, not convinced that the assay effectively scores aggressive behavior.

Line 179 – A relatively recent study suggests zebrafish larvae can see and show negative phototaxic responses near the infrared light spectrum (see Hartman et al. (2018) “Zebrafish larvae show negative phototaxis to near-infrared light”). Because this plays an integral part in the study, authors need to specify the wavelengths of infrared light used. This could have a significant impact on the sleep behaviors observed.

Line 193 – Were the tracks of the zebrafish edited and screened for errors?

Line 321 -  GABA release was not quantified in any way in this study, how can you make this conclusion?

Line 332-334 – This is repetitive and reads like the results section

Author Response

Reviewer  2:

Specific Comments:

  1. Line 57 – the function of scn1lab should be described to provide context;

- We added this ”Mutations of SCN1A gene responsible with voltage-gated sodium channels functioning were linked to neurologic disorders [31].” beside ” The homozygous scn1labs552 zebrafish mutants does not exhibit night time hyperactivity, or reduced responsiveness to dark after 25 µM and 250 µM VPA administration [31].”

  1. Line 147 – with only 30s of acclimation could the ‘novel tank’ not impact the behavioral response? How did you control for the novel tank effect?

- We did some trials before applying the social interaction and aggression tests before starting the effective study. This was done for discovering if there are tracking errors, to see how the fish will behave in that maze and if 30 s are sufficient for acclimation. We, also, saw in the literature this period and we used it in another experiments; these 30s being enough for fish acclimation.

  1. Line 164 - How did you distinguish between aggressive and social behaviors in the mirror? Being in the same zone as the mirror does not necessarily indicate an aggression response, not convinced that the assay effectively scores aggressive behavior.

- The mirror stimulus is the most common method used in evaluating the aggression of zebrafish and it is more used to assess aggression than sociability. We observed the behavior of each fish during the aggression test and we saw darts in the case of Group 4, beside the amount of time spent near mirror.

  1. Line 179 – A relatively recent study suggests zebrafish larvae can see and show negative phototaxic responses near the infrared light spectrum (see Hartman et al. (2018) “Zebrafish larvae show negative phototaxis to near-infrared light”). Because this plays an integral part in the study, authors need to specify the wavelengths of infrared light used. This could have a significant impact on the sleep behaviors observed.

 - The wavelength of the infrared light used in this study was 850 nm.

  1. Line 193 – Were the tracks of the zebrafish edited and screened for errors?

- We used the software EthoVision Xt which is verified every time before starting recording and checked trials are made to be sure that the results will be correct. In addition, before analyzing the trials we apply a filter in EthoVision Xt which removes any trial that has not been successfully tracked for at least 90 % of the time to avoid the inclusion of the dead larvae. Also, the tracks are not edited or screened for errors.

  1. Line 321 -  GABA release was not quantified in any way in this study, how can you make this conclusion?

- There are numerous studies which correlate GABA impairments with ASD. We suggested that administration of certain chemicals (eg. VPA) can lead to GABA dysregulations and we also mentioned that we need more studies to observe if our supposition is true or not.

  1. Line 332-334 – This is repetitive and reads like the results section

- We reformulated those phrases as:”Swimming activity parameters recorded increases in its values for zebrafish larvae with the highest VPA-exposure period.”

Round 2

Reviewer 1 Report

Authors have answered many of my concerns; but I think additional explanation is required for two of my concerns.

  1. My 6th concern was “In Figure 5 it appears maximum movement was observed at 12:00-1:00 pm duration. Even morning light flash at 8:00 am fails to restore the same level of movement. Can the author explain the reason behind this? Also, was food served during this 24 hr observation? If yes how were the movement during those intervals”

The Authors responded with “The larvae were not fed until 7 dpf; therefore, alterations in movement due to the feeding is not a factor”. However, they did not explain why maximum movement was observed at 12:00-1:00 pm duration.

  1. My 7th concern was “In figure 7 Group 4 appears to spend less time compared to Group 1 in both active as well as stationary state. Since the fish of all group were observed for same amount of time, is there any other state apart from active and stationary where Group 4 fish are spending time” Can the author explain what is “active state” and “stationary state” in the Figure description? Was the Figure8 based on only time spent next to the group zone? If yes, that should be reflected in the description.

Reviewer 2 Report

The authors only selectively addressed my comments. The authors need to discuss the possibility of 'tank effect' from inadequate acclimation periods and how this could impact the results. The authors also framed their entire discussion around endpoints that were not the topic of their study, and this needs to be re-written.